# Did Homocysteine Take Part in the Start of the Synthesis of Peptides on the Early Earth?

**DOI:** 10.3390/biom12040555

**Published:** 2022-04-08

**Authors:** Sparta Youssef-Saliba, Anne Milet, Yannick Vallée

**Affiliations:** University Grenoble Alpes, CNRS, DCM, F-38058 Grenoble, France; sparta_saliba@hotmail.com (S.Y.-S.); anne.milet@univ-grenoble-alpes.fr (A.M.)

**Keywords:** prebiotic chemistry, abiogenesis, homocysteine, thiolactone, peptides

## Abstract

Unlike its shorter analog, cysteine, and its methylated derivative, methionine, homocysteine is not today a proteinogenic amino acid. However, this thiol containing amino acid is capable of forming an activated species intramolecularly. Its thiolactone could have made it an interesting molecular building block at the origin of life on Earth. Here we study the cyclization of homocysteine in water and show theoretically and experimentally that in an acidic medium the proportion of thiolactone is significant. This thiolactone easily reacts with amino acids to form dipeptides. We envision that these reactions may help interpret why a methionine residue is introduced at the start of all protein synthesis.

## 1. Introduction

How life came to be on Earth remains a mystery. However, the large number of relevant results published to date allow for some plausible hypotheses [1]. As the first steps that would eventually lead to life were taking place, that is ca. four billion years ago, the main source of carbon on our planet was carbon dioxide [2]. It is thus reasonable to assume that most (if not all) of today’s organic carbon comes from this initial CO_2_ [3]. However, other sources of carbon have been proposed, particularly cyanides [4]. Furthermore, it is possible that some of the earliest bricks of life were not made on Earth but delivered from space at a time when our planet was still undergoing massive meteorite bombardment [5].

Although it does not participate in building proteins, homocysteine **1** is an important amino acid in the current metabolism. Its biochemistry is intimately linked to those of the two proteionogenic sulfur amino acids, cysteine and methionine [6]. When it is mistakenly introduced into the process of synthesizing a peptide in the ribosome, it is rejected by an editing mechanism that produces its thiolactone **2** (Figure 1) [7].

This editing is important since it avoids the synthesis of proteins which would have abnormal behaviors, potentially leading to dysfunctions and to serious diseases [8]. However, in absolute terms, nothing prevents imagining proteins in which all or part of the Cys residues would have been replaced by Hcy residues. The thiol function of homocysteine might play a role similar to that of its shorter counterpart. Hcy residues could form disulfide bridges, enter in the composition of metal complexes and lead to some radical chemistry. However, although plausible prebiotic syntheses of this amino acid have been reported [9,10], for some reason, probably at an early stage of its development, life rejected the possibility to introduce it in proteins.

If, due to the specificity of their thiol function, Cys residues play a considerable role in today’s proteins, especially in multiple catalytic processes, Met residues can generally be considered as lipophilic residues little different from others (Leu, Ile, Val...) and easily interchangeable with them, although sometimes their sulfur atom chelates a metal [11]. No doubt, the most striking role of Met is that it starts the synthesis of all archaea and eukaryotes proteins, and that its *N*-formylated derivative starts the synthesis of all bacteria proteins. It has been proposed that this choice, if not performed purely at random, could be correlated with the relative rarity of the Met residue (2.32% in a representative set of proteins [12]) and with the fact that it is encoded only by a single codon (AUG). Compared to a more common residue and/or a residue encoded by several codons, this would be a better indication to signify a start residue. A notable difficulty here is that the consensus on the order of introduction of the 20 usual residues in the genetic code ranks methionine as one of the very last amino acids to enter the code [13]. There would therefore have been a period with another start residue or without a defined start residue.

In our genetic world, the real starting point is not so much Met but its codon, AUG. However, many exceptions to the use of AUG are known. For example, 17% of the start codons in *Escherichia coli* are different from AUG (mostly GUG, UUG) [14,15]. However, independent of the used start codon, fMet always starts the synthesis of peptides. That is, fMet looks more invariable than AUG. This variability of the start codon may be the result of late adaptations in the history of life. Alternatively, it could mean that Met (or fMet) was chosen before the start codon, which then should have adapted to the presence of the start Met. However, nothing in the chemistry of methionine predisposes it to be more reactive than other amino acids. This is not the case with its demethylated analog, homocysteine, which can form its thiolactone. Such an activated species could have played an important role in the thioester world proposed by de Duve [16].

In this article, we study the cyclization in water of homocysteine into its thiolactone, first experimentally, then using theoretical calculations. We also present our results about the reaction of this thiolactone and of two of its *N*-acylated derivatives with amino acids to form dipeptides.

## 2. Materials and Methods

^1^H-NMR and ^13^C-NMR spectra were recorded on a Bruker Avance 500 (Billerica, MA, USA) (1H:500 MHz, 13C:125 MHz) and 400 (1H:400 MHz, 13C:100 MHz) spectrometers.

High-resolution mass spectra (HRMS) were recorded on LTQ Orbitrap XL (Thermo Scientific, Illkirch, France) mass spectrometer.

Homocysteine **1**, thiolactone **2** (as its HCl salt), *N*-acetylthiolactone **6** and all used amino acids are commercially available. They were purchased and used without further purification.

Synthesis of N-formyl thiolactone **4** [17]. Acetic anhydride (60 mL) was mixed with 30 mL of formic acid under inert atmosphere. The solution was stirred at room temperature for 24 h. A total of 10 g of homocysteine thiolactone was added and the reaction left to stir under reflux for 5 h. An oil was obtained after vacuum elimination of the solvents. This oil was purified by flash column chromatography (90:10 ethyl acetate:pentane); **4** was obtained in a 68% yield.

^1^H-NMR (500 MHz, CDCl_3_):8.19 (s, 1H, C(O)H), 6.57 (s, 1H, NH), 4.55 (m, 1H), 3.27 (m, 2H, CH_2_), 2.85 (m, 1H of CH_2_), 1.93 (m, 1H of CH_2_). ^13^C-NMR (125 MHz, CDCl_3_):205.2, 161.8, 58.1, 31.7, 27.6. HRMS (ESI):Calcd for C_5_H_8_O_2_NS [M + H]^+^:*m*/*z* = 146.02703, found:*m*/*z* = 146.02669.

Most reactions were run in NMR tubes with no purification of products. The obtained products were identified in reaction mixtures. Only in the case of glycine, preparative reactions were run which permitted the isolation of formed dipeptides.

**2** + glycine. Homocysteine thiolactone **2** HCl salt (5 mg, 0.033 mmol) was mixed with 2 equivalent of glycine (4.9 mg, 0.066 mmol) in 5 mL of degazed distilled water. The solution was basified to pH 8 by adding NaHCO_3_. The mixture was stirred at 45 °C for 24 h. The crude product was purified on an inverse phase silica preparative column (50:50:1 H_2_O:ACN:TFA). Hcy-Gly was isolated in a 26% yield.

^1^H-NMR (500 MHz, D_2_O):4.07 (dd, *J* = 4, 7 Hz, 1H), 3.75 (dd of AB system, *J* = 17, 30 Hz, 2H), 2.82 – 2.71 (m, 2H), 2.28 – 2.23 (m, 2H). ^13^C-NMR (125 MHz, D_2_O):175.99, 168.98, 52.52, 43.39, 31.80, 29.99. HRMS (ESI):Calcd for C_12_H_21_N_4_O_6_S_2_ (disulfide) [M−H]:*m*/*z* = 381.09080, found:*m*/*z* = 381.09088.

**4** + glycine. Similarly, we isolated *N*-formyl-Hcy-Gly, from the reaction of **4** with glycine (yield 56%).

^1^H-NMR (500 MHz, D_2_O):8.10 (s, 1H, C(O)H), 4.56 (dd, *J* = 6, 9 Hz, 1H), 3.68 (m, 2H), 2.54 (m, 2H, CH_2_), 1.98 (m, 2H, CH_2_). ^13^C-NMR (125 MHz, D_2_O): 177.3, 161.2, 59.2, 40.5, 32.4, 28.2. HRMS (ESI):Calcd for C_7_H_11_N_2_O_4_S [M−H]^−^:*m*/*z* = 219.04450, found: *m*/*z* = 219.04460.

**6** + glycine. *N*-acetyl-Hcy-Gly was isolated in a 46% yield. ^1^H-NMR (500 MHz, D_2_O):4.42 (dd, *J* = 6, 9 Hz, 1H), 3.92 (m, 2H), 2.82–2.65 (m, 2H), 2.21–1.97 (m, 2H), 1.95 (s, 3H). ^13^C-NMR (125 MHz, D_2_O):174.36, 174.12, 172.93, 52.36, 41.04, 33.74, 30.46, 21.71. HRMS (ESI):Calcd for C_16_H_25_N_4_O_8_S_2_ (disulfide) [M−H]^−^:*m*/*z* = 465.11193, found: *m/z* = 465.11168.

Dimerization of **2** on silica. Homocysteine **1** (5 mg) was diluted in a minimum amount of water (less than 0.1 mL). An excess of silica was added to the solution. Water was then evaporated under vacuum with heating at 60 °C. The obtained dry solid was then left overnight at 80 °C. D_2_O was then added, and the mixture was filtered. The D_2_O phase was collected and analyzed by ^1^H NMR. It was consistent with a mixture 1/2/3 in the ratio 30/45/25. Mass spectrometry of the mixture also confirmed the formation of **3**:HRMS (ESI):Calcd for C_8_H_15_N_2_O_2_S_2_ [M−H]^+^:*m*/*z* = 235.05695, found: *m*/*z* = 235.05702.

## 3. Results

### 3.1. Cyclisation of Homocysteine: Experimental Results

We tested the behavior of **1** in water at various pH values and temperatures. Results are presented in Figure 1. Ratios were estimated using ^1^H NMR spectroscopy. An example of the obtained spectrums is presented in Figure 2.

As observed in Figure 1c, at 80 °C, under very acidic conditions (pH 1), the dominant species is the thiolactone **2** (ratio **1**/**2**: 15/85). At this same temperature, under less acidic conditions (pH 2 to 5), when the equilibrium is reached, the amount of **2** is still important (ca. 15-20%). Only at nearly neutral pH does the proportion of **2** become smaller (a few percent). Results at lower temperatures (45 and 25 °C) are similar.

To mimic primitive conditions that may have existed, for instance, on warm beaches or near volcanoes, we also tested drying conditions [18]. A priori, such conditions should be favorable for water elimination reactions. Water was evaporated from an aqueous solution of **1** over silica, and the resulting dry solid was left at 80 °C overnight. The water-soluble content of this solid was recovered in D_2_O. The ^1^H NMR of the recovered mixture showed it to consist of a mixture of **1**, **2** and piperazinedione **3** [19] (Figure 2). The **1**/**2**/**3** ratio was 30/45/25. Piperazinedione **3**, the cyclic form of HcyHcy, is a dimer of **1**, whose formation can be explained only via the opening of **2**. Thus, **2** can form not only in acidic water but also on a dry surface. As it was obtained here with a high concentration, two molecules of **2** reacted together, which means that the thiolactone underwent amide bond formation, **3** being a cyclic dipeptide.

Additionally, when we dissolved preformed **2** in water at pH 7, its hydrolysis was slow even at 80 °C (less than 1% hydrolysis after **2** weeks). Only under basic conditions was the opening of **2** quicker (60% hydrolysis after 24 h at 25 °C).

### 3.2. Formation of Dipeptides from Homocysteine Thiolactone and Its N-formyl and N-acetyl Derivatives

We then tested the reactivity of **2** with amino acid. All reactions were run at 45 °C. Representative experiments are summarized in Figure 3 (other examples, including examples with small peptides, are presented in Appendix A). At pH values lower than 7, no reaction was observed. At pH 7 (fixed by the addition of imidazole to the reaction mixture) and 8 (NaHCO_3_) all tested reactions were efficient, with conversions to dipeptides being a little better at pH 8.

We also tested the reactivity of the formylated thiolactone **4** with glycine (Figure 4). The reaction was run at 45 °C, pH 8, for 24 h, after which time the ^1^H NMR of the reaction mixture clearly indicated the formation of **5**.

Similarly, *N*-acetyl homocysteine
thiolactone **6** reacted with amino acids. Representative examples are presented in Figure 5 (other examples in Appendix A).

## 4. Discussion

### 4.1. Cyclisation of Homocysteine: Theoretical Calculations

Using DFT calculations, both static and dynamic, mechanical insights into the cyclisation can be proposed. All the static calculations were performed using Gaussian 16 [20] at the B3LYP level with the def2-TZVP basis set [21] and solvent effects using the SMD (Solvent Model Density) [22] option to mimic the bulk effect of water. Dispersion was added through the D3 version of Grimme with Becke–Johnson damping. The comparison of the results with and without D3BJ is given in the Appendix A [23,24,25]. All the optimizations were followed by a frequency calculation to check the nature (minimum or transition state) of the optimized geometry. We also added explicit water molecules. For example, three water molecules are located around the NH3^+^ group.

The global energetical cost was computed using the reaction given on Figure 3. The reaction is slightly endo-energetic by around +3 kcal·mol^−1^ (See Appendix A for more information and different levels). However, if we used H_3_O^+^-H_2_O, around the carboxylic acid in Figure 3 instead of two water molecules, the reaction is slightly exothermic with a ∆H of around −7.35 kcal·mol^−1^, which is coherent with a reaction more efficient in acidic condition.

To clarify the discussion of the mechanism, it is important to specify some geometrical characteristics of the reactant and the product. For the reactant, the length of the SH bond was around 1.35 Å, the final C-S bond length was around 1.80 Å against 3.3 Å in the starting nonlinear conformer. A picture of the dynamics of **1** in water and of the surrounding hydrogen bonds network can also be useful. Therefore, we used a General Amber Force Field (GAFF) [26], using Amber18 [27] to describe **1** and run a dynamic of **1** plus 2410 water molecules and one chlorine anion for neutrality for 70 ns after a process of equilibration (see Appendix A for details). The trajectories were analyzed using cpptraj, the program-processing amber trajectories files [28]. We then extracted data on hydrogen bonds and distances. To define the existence of a hydrogen bond, the default values of cpptraj were always used: i.e., the distance cutoff from acceptor to donor heavy atom was 3.00 Å and the angle cutoff was set to 135°. For the hydrogen bond surrounding the SH group, a cutoff of 3.4 Å was used and the angle cutoff maintained at 135°. The OH group of the carboxylic acid function was then involved as a donor of hydrogen bond with water 96% of the time of the simulation with an average distance between the two oxygens involved of 2.687 Å and an average angle of 163°. The three hydrogens of the NH_3_^+^ group were also involved in a hydrogen bond 74% of the time of the simulation with an average distance of 2.843 Å and an average angle of 157°. The thiol function acted as a donor of hydrogen bonds toward water 58% of the time of the simulation with an average sulfur-oxygen bond distance of 3.199 Å and an average angle of 154°. Then the oxygen of the carbonyl function of the carboxylic group was an acceptor of hydrogen bonds 52% of the time of the simulation (2.809 Å and 156° for the average characteristics). The oxygen of the OH function of the carboxylic acid acted as an acceptor of hydrogen bonds 20% of the time of the simulation (2.841 Å and 155°). The carbon–sulfur bond to be made during the cyclisation ranged from 3 to 6 Å. The histogram from Figure 6 shows that the C-S bond distance is indeed very short, between 3 and 3.5 Å, 5% of the time of the simulation, and 20% of the time the C-S bond distance ranged from 3 to 4 Å. The bond distance reached its highest probability between 4.5 and 5 Å, a range of distance observed nearly 40% of the time during the simulation. These data are shown on Figure 6 and confirm that the folded conformation needed for the cyclisation of **1** is likely in water.

Due to the water network and the different proton transfers involved in the reaction, we decided to use QM/MM (quantum mechanics/molecular mechanics) calculations and metadynamics [29] using the CP2K software [30]. The setup of the system is described in the Appendix A. The QM part consists of the organic molecule plus the surrounding water molecules at a distance of around 6 Å, i.e., 79 water molecules including a hydronium cation. Using this tailored quantum part, made up of the organic molecule and a reasonable number of water molecules around it, gives the protons the opportunity to move from homocysteine 1 to the solvent while remaining close to 1. The objective is to keep the “acidic” aspect of the solution close to the molecule. Metadynamics was used to overcome the problem of observing rare events in conventional molecular dynamics and of finding the reaction coordinate. A series of small repulsive Gaussian potentials centered on the values of some collective variables (CV) were added during the dynamics, preventing the system from revisiting the same points in configurational space and creating a history-dependent multidimensional-biasing potential. A time step of 0.5 fs was used for the dynamics, and the hills of 1 kcal·mol^−1^ were added every 20 fs.

To study this reactivity, three collective variables were used. The first one was the carbon sulfur bond to be formed; the second was the S-H to be broken; the C-O bond was the last variable. The width of the Gaussian used was 0.3 bohr, and a reflective wall at 6.2 bohr was used to prevent sampling the C-S bond at large values, which is of no interest since we study the cyclisation. The observed reaction was the creation of the C-S bond concomitant to the proton abstraction from the S-H bond. The C-O bond was not broken. However, its elongation led to its protonation, and a diol intermediate was then identified (Figure 4). The barrier needed is around 25 kcal·mol^−1^. The joint formation of the CS bond and the breaking of the SH bond during the metadynamics can be seen on Figure 7. The limitation of the CS bond to 3.5 Å is due to the addition of a wall to prevent the useless sampling of these conformations during the metadynamics.

Returning to the static calculation at the B3LYP/def2TZP level, the diol (Figure 4) plus two water molecules and with the three water molecules around NH_3_^+^ lies 12.5 kcal·mol^−1^ higher in energy (∆G of 16.0 kcal·mol^−1)^ than the starting molecule. A proton transfer can occur using the two water molecules as a proton relay leading to the final cyclic molecule. The barrier needed is of 15.1 kcal·mol^−1^ (∆G^‡^ of 14.1 kcal·mol^−1^).

Thus, we propose a two-steps mechanism, with the first leading to a diol 12.5 kcal·mol^−1^ higher in energy than homocysteine with a barrier of 25 kcal·mol^−1^. Then, through a barrier of 15 kcal·mol^−1^, the diol leads to the thiolactone.

### 4.2. Hypothesis: Hcy Was the First StartResidue

We have recently reported that in a “nitrile world” (cyanosulfidic world [4]) **2** was a probable intermediate in the synthesis of homocysteine **1** itself [11]. From what is reported above (both experimentally and theoretically), it is now clear that even not using nitriles, if **1** was present, then **2** was also. Highly acidic conditions are particularly favorable for the formation of **2**. Such conditions could have been present on the early Earth, for instance near black smokers and volcanoes. Under such conditions the cyclized form **2** indeed dominated the linear form **1**. However, even under less acidic conditions, **2** was present in a reasonable proportion. Due to the huge amount of CO_2_ present in the atmosphere, the primitive global ocean was probably slightly acidic [31], meaning that in this ocean, the ratio between **1** and **2** was ca. 5 to 1. Even under nearly neutral conditions, at least if the temperature was high enough (which was probably the case [31]), some quantities of **2** were present. Furthermore, we have noticed that, once formed, **2** is stable in neutral water, even at high temperatures. This is consistent with the known stability of **2** in human serum [32]. Of course, under basic conditions, its hydrolysis is faster. However, its half-life at pH ca. 8 remains non-negligible. We have shown that such pH conditions (from 7 to 8) are necessary to obtain good yields of dipeptides from the reaction of **2** with amino acids.

From these results it appears that the prebiotic synthesis of homocyteine dipeptides would require a pH gradient from acidic (maybe even strongly acidic) for the formation of **2** to neutral or slightly basic for its opening by an amino acid. The pH gradients are of major importance in biology, and they are quite usual in nature, for instance near volcanoes or black smokers [33,34,35]. There is therfore no particular difficulty in postulating their use in prebiotic chemistry [36,37,38].

It is thus reasonable to postulate the existence of Hcy-AA dipeptides on the early Earth (where AA stands for any amino acid), and as homocysteine was able to self-activate (forming its thiolactone), these dipeptides could well have been overrepresented compared to the other possible AA_1_-AA_2_ pairs (for which a necessarily slower inter-molecular activation of AA_1_ was required) [39,40,41]. We hypothesize that these overepresented Hcy-AA interacted with RNAs (proto-tRNAs) to form Hcy-AA-RNAs, and that the presence of a Met residue at the start of the synthesis of all actual proteins is a consequence of these primitive reactions. If so, the very first “start residue” was not Met but Hcy, and it was not coded: there was no starting codon. It is only later, when homocysteine was excluded from the list of the 20 canonical amino acids (probably because its presence was detrimental to the stability and to the properties of proteins [42]) that it was replaced as first residue by its *S*-methylated analogue, methionine, which implied the introduction of AUG as start codon. The fact that AUG is not always the codon for the start residue Met can then be regarded as a consequence (and a remembrance) of this late substitution of Hcy by Met, with the need to introduce a start codon corresponding to Met where there was none before. In any case, this would be consistent with the fact that an amino acid that was introduced so lately in the genetic code could still be chosen as the start residue. Moreover, the presence of Hcy residues elsewhere in peptides would have allowed the development of the typical reactivity of thiols in addition to that of Cys residues. Even thiolactone **2** could have been involved in varied radical processes [43,44].

## Data Availability

Not applicable.

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
