# Peer review of "Did Homocysteine Take Part in the Start of the Synthesis of Peptides on the Early Earth?"

_biomolecules, 2022, doi:10.3390/biom12040555_

Round 1

Reviewer 1 Report

The manuscript by Youssef-Saliba et al. describes the behavior of homocysteine under plausible prebiotic conditions, including its reactivity for intramolecular thiolactone formation and peptide formation. The manuscript is straight-forward, clear and thorough. I will add the caveat that I am not an expert in the computational aspects of the work and thus will abstain from commenting on that portion of the work. For the rest of the results presented, however, I believe the manuscript can be accepted after very minor revisions are made. I will describe these below:

For Figure 3, 5 (and all figures describing reactions), it would be helpful (at least for me) to include the structures of the compounds in addition to the number and description (eg. 2+glycine) with the figure so that the reader does not have to refer to the text to remember what #2 is.

Figure 4 is rather low resolution and should be updated with a clearer image.

The DFT calculations are not discussed at all in the results section, only in the discussion. There should be some discussion of both.

Author Response

Reviewer 1

We thank the reviewer for his/her opinion on our paper and his/her comments.

# For Figure 3, 5 (and all figures describing reactions), it would be helpful (at least for me) to include the structures of the compounds in addition to the number and description (eg. 2+glycine) with the figure so that the reader does not have to refer to the text to remember what #2 is.

We have done it. See Fig 3 and 5

# Figure 4 is rather low resolution and should be updated with a clearer image.

We tried to ameliorate it.

# The DFT calculations are not discussed at all in the results section, only in the discussion. There should be some discussion of both.

It would be possible. However, we think that the theoretical part is more readable as it is presented (in only one part). We left it that way. Of course, if it is mandatory in the opinion of the reviewer, we will be pleased to modify it.

Reviewer 2 Report

The manuscript entitled 'Did homocysteine took part in the start of the synthesis of peptides on the early Earth?' by Vallée and co-workers investigates the function and potential of homocystein as building block in the context of the origin of life. Most remarkable is, that homocysteine can form the thiolactone, which might have interesting properties.

Here are my comments:

p.2 l. 45: Hcy residues could form 43

disulfide bridges, enter in the composition of metal complexes, lead to some radical 44

chemistry... However... There seems to miss a part of the sentence.

Concerning the formation of peptides under prebiotic conditions I am missing the following references:

  1. S. Foden, S. Islam, C.  Fernández-García, L.  Maugeri, T. D.  Sheppard, M. W.  Powner, Science 2020, 370, 865.

Prebiotic synthesis of cysteine peptides that catalyze peptide ligation in neutral water

  1. Canavelli, S. Islam, M. W.  Powner, Nature 2019, 571, 546-549.

Peptide ligation by chemoselective aminonitrile coupling in water

  1. Sauer, M. Haas, C.  Sydow, A. F.  Siegle, C. A.  Lauer, O.  Trapp, Nature Communications 2021, 12, 7182.

From amino acid mixtures to peptides in liquid sulphur dioxide on early Earth

Thiolactone is an interesting molecule and there is a potential for radical reactions or even better photoredox organocatalysis or activation. Please add the following references, which discuss the potential for such important processes:

  1. Fuks, L. Huber, T.  Schinkel, O.  Trapp, Eur. J. Org. Chem. 2020, 6192-6198.

Investigation of Straightforward, Photoinduced Alkylations of Electron-Rich Heterocompounds with Electron-Deficient Alkyl Bromides in the Sole Presence of 2,6-Lutidine

  1. C. Closs, M. Bechtel, O.  Trapp, Angew. Chem. Int. Ed. 2022, 61, e202112563.

Dynamic Exchange of Substituents in a Prebiotic Organocatalyst: Initial Steps towards an Evolutionary System

The quality of the manuscript could be improved by adding some kinetic analysis of the reaction progress profiles.

Just for the curiosity: Do you see oxidative coupling? Maybe you can comment on this in the manuscript.

Please correct some typos in the experimental part (Ethyl Acetate: Pentane// ethyl acetate: pentane (n-pentane?)

Author Response

Reviewer 2

We thank the reviewer for his/her opinion on our paper and his/her comments.

# p.2 l. 45: Hcy residues could form 43 disulfide bridges, enter in the composition of metal complexes, lead to some radical 44 chemistry... However... There seems to miss a part of the sentence.

We modified the sentence to make it clearer.

# Concerning the formation of peptides under prebiotic conditions I am missing three references.

The three references have been added (refs 39, 40, 41)

# Thiolactone is an interesting molecule and there is a potential for radical reactions or even better photoredox organocatalysis or activation. Please add two references.

The two references have been added (refs 43, 44).

# The quality of the manuscript could be improved by adding some kinetic analysis of the reaction progress profiles.

Sorry, we have not done that. Our idea was to present a qualitative vision of these reactions: They are possible at reasonable temperatures, with acceptable conversions for relatively short periods of time.

# Just for the curiosity: Do you see oxidative coupling? Maybe you can comment on this in the manuscript.

No. The process is redox neutral. Of course after long period of times (especially at high temperature) small amounts of disulfides were detected in some experiments. But this was due to some O­2 entrance in the reaction vessel (despite of course reactions were run under neutral atmosphere).

# Please correct some typos in the experimental part (Ethyl Acetate: Pentane// ethyl acetate: pentane (n-pentane?)

Done